# Fall armyworm infestation, maize production and nutrition security: Evidence from Uganda

Brian Chisanga[1,2]*, Menale Kassie[2]

**1** Development Economics Group, Wageningen University and Research, the Netherlands, **2** International Centre of Insect Physiology and Ecology (ICIPE), Nairobi, Kenya

* brian.chisanga@wur.nl

## Abstract

We study the impact of fall armyworm (FAW) infestation on nutrition security outcomes in eastern Uganda, measuring nutrition security by dietary diversity scores of vulnerable household members—children under 5 years and their mothers. We use different regression models and aim to take the endogeneity of FAW infestation seriously. We also analyse FAW's impact pathway to nutrition status and ask whether impacts are caused by reduced maize yields and sales or increased costs associated with insecticide use. The main results are that high FAW infestation reduces maize yields and sales, and adversely affects dietary diversity.

## Introduction

Rapid diffusion of the fall armyworm (FAW, *Spodoptera frugiperda*) across the African continent has far-reaching consequences for Africa's agriculture and food security situation. The pest mainly invades maize—the primary staple crop for 300 million people in Africa [1]. It is native to tropical and sub-tropical America [2], but appeared in West Africa in 2016. Since then, it has spread rapidly and has developed into a major pest that causes significant crop damage and economic losses. Current estimates indicate annual losses of 4–18 million tons of maize due to FAW [(e.g., 3)], valued at US$2.5–6 billion. This quantity of maize could feed 40 million to 100 million people [4,5]. FAW attacks the maize crop at all stages of its growth, but usually during early stages with potential to cause total crop failure [1,6,7]. Loss rates of 20–50% in major production regions are not uncommon [4,8,9].

In this paper we study the impact of FAW infestation on nutrition security outcomes in eastern Uganda. We use Dietary Diversity Scores (DDS) and focus on the most vulnerable household members—children under the age of 5 years and their mothers. We also wish to understand FAW's impact pathway to nutrition status. While the prevalence of FAW risk is likely exogenous to individual farms [10–13], we take into account that the "intensity of infestation" may be endogenous and affected by farm management choices. Thus, we test for the endogeneity of our treatment variables,

OPEN ACCESS

**Data availability statement:** The data underlying the results presented in the study are available from Zenodo repository: https://zenodo.org/records/17229629 and DOI: https://doi.org/10.5281/zenodo.17229629.

**Funding:** Financial support for this research by the following organizations and agencies: European Union (DCI-FOOD/2018/402-634). Funders had no role in study design, data collection and analysis, decision to publish, or preparation of the manuscript.

**Competing interests:** The authors have declared that no competing interests exist.

the "intensity of infestation," before estimating treatment effects. We estimate both Ordinary Least Squares (OLS) models and a Multinomial Endogenous switching regression (MESR). The empirical analysis is based on survey data from a region in Uganda collected during the 2018/2019 production season.

In Uganda, approximately 70% of maize is produced by smallholder farmers—a category that may be particularly vulnerable to the adverse effects of FAW [14,15]. Smallholder farmers' welfare depends on maize production as this crop contributes to their food consumption and income (via sales of surplus maize). About 40 percent of Uganda's caloric requirements are from maize, with estimated consumption levels of 22 kgs per capita annually [15,16]. If FAW invasions reduce maize production, farmers' consumption is at risk. FAW may also limits dietary diversity because reduced maize sales will limit opportunities to purchase food items on the market. Affected households relying on markets for part of their consumption basket may therefore, consume less diverse diets. Another channel linking FAW infestation to nutrition outcomes is increased pesticide expenditures to curb maize production losses due to FAW—pesticides are expensive in our study region. These expenditures may crowd out food expenditures. This theory of change is summarized in Appendix Figure A1 in S2 Appendix.

Our main findings are as follows. First, echoing other literature, we document that FAW infestation is associated with significant declines in maize yields and sales. However, we find that low FAW intensity has no effect on maize yield, while high FAW infestation causes substantial yield losses. These impacts are mirrored in our findings on nutrition status where we document no effect of low FAW intensity on dietary diversity but find a sizeable impact of high FAW on both mothers' and children's diets. Maize sales suffer more profound declines resulting from both low and high FAW intensity with high FAW infestation resulting in larger sales decline than low infestation rates. Our results further show that pesticides-use increases with low FAW intensity and yet high FAW infestation has no effect on pesticide-use. While this finding is counter-intuitive, it could reflect a lack of liquidity or an unwillingness by farmers to apply costly insecticides on crops that they believe cannot be salvaged anyway. Importantly, we find that FAW causes significant and negative effects on the diets of mothers and children—reducing their dietary diversity. Our finding that the impact is only found at high FAW intensity, signifies that household diets are rather resilient to FAW at low exposure. High FAW intensity has a larger impact on mothers than children. Notwithstanding, we emphasize that the nutrition security status of this population is precarious even in the absence of FAW. Nearly half the population is "food-energy deficient" [17]. Stunting rates for children under five are 29% [18], and the prevalence of underweight women of reproductive age in Uganda is estimated at 6.9% [19].

This paper extends the literature in various ways. Following Kassie et al. (2020), but extending most of the literature, we aim to consider the endogeneity of FAW infestation intensity that varies with farm management choices—even if the occurrence of pests is likely a matter of bad luck. Hence, we conduct various endogeneity tests to confirm which model is appropriate between OLS or the MESR, which

accounts for endogeneity and create a counterfactual scenario. Unlike Kassie et al. (2020), our analysis extends beyond productivity and marketing outcomes and zooms in on nutritional security outcomes. Tambo, Kansiime [20] and Bannor, Oppong-Kyeremeh [11] also study the impact of FAW prevalence (binary treatment) on food security, but their measure of food security is the Household Food Insecurity Access Scale (HFIAS), which measures consumption effects of FAW rather than effects on diet quality. The HFIAS focuses on calories from starch and therefore may mask potential nutritional deficiencies. Moreover, we use individual dietary diversity scores rather than an aggregate household-level variable as our dependent variables. This distinction is relevant if the intra-household distribution of resources is unequal, and does more justice to outcomes of the most vulnerable household members [21–23].

This paper is organized as follows. In the materials and methods section, we discuss the econometrics approach. This is followed by the data and descriptive statistics, which introduces and summarizes our data and key variables. We presents the main regression results in the results and discussion section, and the paper closes with a conclusions section.

## Materials and methods

### Econometric approach

We expect that FAW infestation will adversely affect the nutrition security of affected farmers and foresee two main pathways. First, there is a production pathway. FAW mainly targets maize, so there are direct impacts via reduced production for own consumption and *indirect* impacts via reduced disposable income because less maize can be sold. Second, there is an expenditure pathway, as farmers may perceive the need to increase pesticide use to curb the intensity of the infestation. This paper explores whether FAW infestation causes less diverse nutritious diets, either by reduced maize sales or increased pesticide expenditures.

We present our estimation strategy to determine how FAW affects our outcome variables. While the presence of FAW on specific plots is likely exogenous [10,11], the extent to which the pest causes crop losses is a function of various factors, including management activities of the farmer. Reflecting that the intensity of FAW infestation may be endogenous we specify a Multinomial Endogenous Switching Regression (MESR) model. We asked households to determine the proportion of maize harvest for each plot that was lost due to the FAW. Based on self-reported assessments, we construct a categorical variable "FAW intensity", taking a value of zero if the plot was not affected by FAW, a value of one indicating "low FAW intensity" (if maize harvest lost was less than 30% of the total harvest). A value of two indicates "high FAW intensity" (if maize harvest lost was more than 30%). We use 30% as the threshold because this is the median FAW intensity in our sample, so we have two equally-sized groups of FAW intensity. Recent studies estimate a mean yield loss of 31%, close to our median [3,4,9,10,24,25]. Also see Appendix Table A.8 in S1 Appendix.

We compare the three regimes of FAW intensity using the MESR model, estimating counterfactual outcomes using conditional expectations while controlling for observed and unobserved heterogeneity [26–28]. In the first step, we estimate FAW intensity as a function of plot and household characteristics using a Multinomial Logit selection (MNLS) model:

$$FAW_{pij} = \boldsymbol{X_{pi}}\beta_j + \varepsilon_{pij}, \tag{1}$$

where $FAW_{pij}$ ($j = 0, 1, 2$), is our categorical variable FAW intensity. The vector $\boldsymbol{X}$ captures plot and household variables affecting FAW intensity; $\beta_j$ are parameters to be estimated for each FAW regime $j$; and $\varepsilon_{pi}$ is a random error term with mean zero and variance $\sigma^2$. FAW intensity can be affected by weather conditions, such as temperature, wind and rainfall [13], and soil characteristics, which are unobserved in our data. These unobserved factors are left as part of the error term in the first step.

In the second step, we specify three separate outcome equations as linear regressions for the three FAW intensity regimes $j$:

$$\begin{cases} Regime\ 1: & Y_{pi0} = \mathbf{Z}_{pio}\delta_0 + \mu_{pi0} & if\ FAW = 0 \\ Regime\ 2: & Y_{pi1} = \mathbf{Z}_{pi1}\delta_1 + \mu_{pi1} & if\ FAW = 1 \\ Regime\ 3: & Y_{pi2} = \mathbf{Z}_{pi2}\delta_2 + \mu_{pi2} & if\ FAW = 2 \end{cases} \qquad (2)$$

where $Y_{pij}(j = 0, 1, 2)$ are outcome variables (maize yield, sales, insecticide use, dietary diversity scores for children under 5 years and mothers); $\mathbf{Z}_{pij}$ is a vector of plot and household variables (including $FAW_{pi}$) that affect outcome variables; $\delta_j$ are parameters to be estimated; and $\mu_{pij}$ are random error terms.

OLS estimates will be biased if error terms $\varepsilon_{pij}$ are correlated with error terms $\mu_{pij}$. The correlation can stem, for instance, from differences in farmers' unobserved behavior or management abilities, intrinsic motivation or business acumen, attention to detail to control FAW and unobserved plot-level characteristics such as soil fertility. For example, poorly managed plots may suffer from significant damage and low productivity. Consistent estimation of $\delta_j$ requires the inclusion of the selection correction term, also known as the Inverse Mills Ratios (IMR). Following Di Falco [28], Kassie, Marenya [29], and [26], the second stage of MESR with consistent estimates is specified as follows:

$$\begin{cases} Regime\ 1: & Y_{pi0} = \mathbf{Z}_{pio}\delta_0 + \sigma_0\hat{\lambda}_{pi0} + \mu_{pi0} & if\ FAW = 0 \\ Regime\ 2: & Y_{pi1} = \mathbf{Z}_{pi1}\delta_1 + \sigma_1\hat{\lambda}_{pi1} + \mu_{pi1} & if\ FAW = 1 \\ Regime\ 3: & Y_{pi2} = \mathbf{Z}_{pi2}\delta_2 + \sigma_2\hat{\lambda}_{pi2} + \mu_{pi2} & if\ FAW = 2 \end{cases} \qquad (3)$$

where $\sigma_j$ is the covariance between the $\mu_{pij}$'s and $\varepsilon_{pij}$'s, and where $\hat{\lambda}_{pij}$ is the IMR computed from estimated probabilities from [1]. The number of bias correction terms in each equation equals the number of multinomial logit choices.

For identification purposes, we need at least one instrumental variable per endogenous variable [26,30]. We develop three instrumental variables for our two endogenous variables (low FAW and high FAW damage). Instruments are in $\mathbf{X}$ but omitted in $\mathbf{Z}$. The first instrument to identify [1] is a household-level variable: "Risk of FAW". This variable captures the local density of FAW and thus proxies for the risk and magnitude of infestation and re-infestation of a household's plot from neighboring fields. We use Principal Component Analysis (PCA) to identify household clusters within 200 meters radius to each other. We identified a total of 163 cluster within our sample. For each household, the instrument is defined as the ratio of FAW-infested plots in the cluster (excluding the farmer's plots) to the total number of plots in the cluster (again excluding the farmer's plots). This assumes that FAW spreads most easily across contiguous or nearby plots. The risk of FAW infestation and re-infestation is highest if the neighbors' plots are affected. In other studies, distance is also used to construct instruments [(see 26,27,31)]. We expect the exclusion restriction to hold because the variable "Risk of FAW" is unlikely to affect our outcome variables through any channel other than the intensity of FAW on the farmer's own plot. The instrument "Risk of FAW" is inspired by, among others, comparable approaches taken by [32,33], who study rural-urban migration and construct a proxy for rural migration networks as the shares of households with seasonal and permanent migrants to the total number of rural households at the community level.

The second instrument we use is the distance between a plot in our study that was not infested with FAW and the closest that was infested. We used geo-referenced data to identify the nearest plot to an unaffected plot among our study plots and then we estimate distance between the plots. This IV proxies the likelihood that a FAW unaffected plot could become infested due to proximity to infested plots. Again, the exclusion restriction here, is that our outcome variables cannot be directly affected by the IV unless through the endogenous variable.

Our third instrument is distance between the household's residence and the plot. This instrument proxies the management of FAW intensity by the household. The proximity of the plot to the household may affect the likelihood of the household to inspect the field regularly and take appropriate measures to manage FAW intensity—which can be labor intensive. The exclusion restriction requires that the instrument only affects the outcome variables through the endogenous variable.

While ideally, we would have used a panel data approach to address unobserved farmer's characteristics, this is unfortunately not possible with our data. Following Manda, Alene [34] we exploit plot-level information to deal with the issue of farmers' unobservable characteristics [(e.g., 35–37)]. This approach to controlling for unobserved heterogeneity is based

on Mundlak [38]. We include on the right-hand side of each equation the mean value of plot-varying explanatory variables. The main assumption of this approach is that unobserved effects are linearly correlated with the means of the plot-varying explanatory variables.

Finally, following previous studies [23,26,34], we derive conditional expectations from the MESR model to compute counterfactual scenarios. The MESR allows us to produce selection-corrected predictions of counterfactual outcomes [28]. We use actual and counterfactual scenarios to compare the expected values of outcomes of high- and low-FAW intensity plots with FAW-unaffected plots, which is the reference group. Counterfactual outcomes refer to the expected outcomes for high- and low-FAW intensity plots if they had not been affected by FAW.

## Estimating the average treatment effects on the treated (ATT)

We compute the average treatment effects on the treated (ATT), indicating the average effect of FAW intensity. ATT is the difference between the actual and counterfactual scenarios. Equation (4) gives the expected outcomes for actual scenarios, i.e., the expected outcome for low-FAW intensity plots given the predicted actual scenario of low intensity ($j = 1$) and the expected outcomes for high-FAW intensity plots given the predicted actual scenario of high FAW intensity ($j = 2$):

$$E\left(Y_{pij}\middle|FAW = j\right) = \mathbf{Z}_{pij}\delta_j + \sigma_j\hat{\lambda}_{pij}, \quad (j = 1, 2)$$
(4)

Instead, equation (5) gives expected outcomes for the counterfactual scenarios. That is, the expected outcome for low-FAW intensity plots under the scenario that they had not been affected by FAW ($j = 0$). This provides estimates of the value of the outcome variables for low-FAW intensity plots ($j = 1$) based on the assumption that the coefficients for $\mathbf{Z}_{pij}$ and $\hat{\lambda}_{pij}$ are the same as for FAW-unaffected plots ($j = 0$). Similarly, for plots affected with high-FAW ($j = 2$), the counterfactual outcome is the expected outcome that would occur under the scenario that they had not been affected with FAW ($j = 0$):

$$E\left(Y_{pi1}\middle|FAW = j\right) = \mathbf{Z}_{pij}\delta_0 + \sigma_0\hat{\lambda}_{pij}, \quad (j = 1, 2)$$
(5)

We combine (4) and (5) to compute the Average Treatment Effect on the Treated (ATT) by taking the difference between expected outcomes for actual and counterfactual scenarios, or:

$$ATT = E\left(Y_{pij}\middle|FAW = j\right) - E\left(Y_{pi1}\middle|FAW = j\right) = ATT = \mathbf{Z}_{pij}\left(\delta_j - \delta_0\right) + \hat{\lambda}_{pij}\left(\sigma_j - \sigma_0\right).$$
(6)

## Decision on the appropriate model between OLS and MESR

Our decision on which model to use between OLS and MESR was based on our tests for the three instruments and for each of the outcome variables. Firstly, we conduct a falsification test following (Di Falco, Veronesi [39], based on the inclusion of the instrument in the outcome equations for plots unaffected by FAW. If the instrument explains selection into infestation but does not enter significantly when explaining outcomes when FAW = 0, then we have some confidence that the exclusion restriction is satisfied. Secondly, we consider the F-statistic from the first stage or reduced-form regression from the Two Stage Least Square (2SLS). As a rule of thumb, an instrument is considered strong if the F-statistic is 10 or greater. Finally, we considered the results of the Durbin-Wu-Hausman (DWH) tests to make the final decision on the appropriate model. The DWH tests for the exogeneity of the potentially endogenous regressors. If the p-value is greater than zero, then the suspected regressor was indeed exogenous and OLS estimates are more appropriate as addressing endogeneity is not necessary.

## Data and descriptive statistics

Our main data source is a household survey conducted in Kamuli District in Eastern Uganda. Kamuli District is in the Busoga sub-region at an altitude of 1,083m above sea level. Its size is 3,443 km$^2$ with a population of more than 500,000 people. We use data collected during the short rainy season in September/October 2018 and the long rainy season in March/April 2019, which together consist of a full agricultural year (and which therefore are pooled as a cross-section). Data was accessed by researchers in June 2020. We used a multi-stage stratified random sampling framework to select 11 sub-counties out of 13 sub-counties in the district, and randomly sampled villages so that the probability of inclusion was proportional to size. Our survey sample consists of 920 randomly selected households who managed 1,042 maize plots in the short rainy season and 1,115 maize plots in the long rainy season. For the short rainy season, farmers reported that 847 out of 1042 maize plots (or 81%) were infested with FAW, and in the long rainy season, 917 maize plots out of 1,115 (82%) were reported to be infested. Pooling observations across the two seasons, 393 plots (18%) were not affected by FAW, 877 (41%) suffered from low FAW intensity, and 884 (41%) suffered from high FAW intensity. Permission for this study was granted by the District Agricultural officer for Kamuli district, who works under the Ministry of Agriculture, Animal Industry and Fisheries (MAAIF).

We now describe our five outcome variables: maize yield, maize sales, insecticide use, a dietary diversity score (DDS) for children, and DDS for mothers. The two DDS are the main outcome variables, and maize yield, sales, and pesticide variables are intermediate variables that we use to identify the mechanism linking FAW infestation to nutrition outcomes. Table 1 summarizes these variables for the three sub-groups separately and for the full sample. Detailed descriptive statistics are summarized in Appendix Table. A1.

The first outcome variable is maize yield measured in kg per hectare for each plot. Not surprisingly, the intensity of FAW infestation is correlated with yields. From columns (0), (1), and (2), non-infested plots had a higher yield (1,714 kg/ha) than plots infested with low intensity (1,631 kg/ha) and high FAW intensity (1,341 kg/ha). The average maize yield in Uganda was about 1600 kg/ha in the 2019 agricultural season [40], which reveals that yields in our sample are comparable to the national average. Columns (4–5) report whether differences between infested and non-infested plots is statistically significant. The next row shows that mean maize sales for households during the agricultural season year amounted to 740 kg, valued at UGX 342,757 (US$ 93). Maize sales were 24% of mean production for a household, which is estimated at 3,080 kg from our data. It is interesting to observe that the difference in maize sold across sub-groups is smaller than the difference in yields. This may reflect that households sell surplus maize, but also that they adjust their own consumption to smooth their maize sales income (needed for necessary expenditures).

**Table 1. Summary statistics for outcome variables.**

| Variables | Level of FAW Intensity | | | | | | |
|---|---|---|---|---|---|---|---|
| | (0)<br>0 = No FAW | (1)<br>1 = Low intensity | (2)<br>2 = High Intensity | (3)<br>Full sample | (4)<br>Diff (1)-(0) | (5)<br>Diff (2)-(0) | (6)<br>Obs |
| Maize yield (kg/ha) | 1714.76 | 1631.39 | 1341.75 | 1527.74 | −83.38 | −373.01*** | 2,154 |
| Maize quantities sold (kg) | 640.43 | 619.03 | 353.71 | 506.42 | −21.40 | −286.72*** | 1,726 |
| Insecticide use (ltr/Ha) | 0.42 | 0.60 | 0.52 | 0.54 | 0.19** | 0.10 | 2,154 |
| DDS Children (Score)-24 hours recall period | 6.71 | 6.46 | 6.17 | 6.34 | −0.26 | −0.54*** | 782 |
| DDS Children (Score)-7days recall period | 8.98 | 9.14 | 9.18 | 9.14 | 0.16 | 0.20 | 782 |
| DDS Mothers (Score)-24 hours recall period | 6.76 | 6.38 | 6.12 | 6.27 | −0.29* | −0.49*** | 1,427 |
| DDS Mothers (Score)-7 days recall period | 8.79 | 8.80 | 9.00 | 8.93 | −0.10 | 0.21 | 1,427 |

*, **, *** significant at 10%, 5% and 1%.

Regarding pesticide application, we expected that households with infested plots were more likely to apply pesticides and use greater quantities at more significant costs. While this is confirmed by the data for plots suffering from low intensity, the same is not true for plots suffering from high FAW intensity. Pesticide application declines as the level of FAW intensity increases beyond a certain threshold value, and households suffering from high pest pressure are less likely to use pesticides and, if they do use pesticides at all, they apply smaller quantities. This may capture expectations about lower crop income and binding cash constraints (as maize sales decline). It may also reflect that some farmers consider heavily infested plots as a "total loss" and give up on them—not throwing good money after bad if they perceive there is little harvest to be salvaged. According to our data, only 32 percent of farmers used improved seed and 20 percent used chemical fertilizers (urea and/or diammonium phosphate (DAP)).

Our main dependent variables are measures of dietary diversity. We compute DDS for children under five and their mothers, using guidelines developed by the Food and Agriculture Organisation (FAO) [41]. We define the DDS as the number of unique food groups consumed in the previous 24-hour period. Our survey captured 22 food groups, which we aggregated into 12 groups, as recommended. The 12 food groups are cereals, white tubers & roots, vegetables, fruits, meat, eggs, fish & other seafood, legumes, nuts & seeds, milk & dairy products, oil & fats, sweets, spices, condiments, and beverages. We use the DDS based on a recall period of 24 hours and treat the main alternative measure (based on a recall period of 7 days) as a robustness check. While the variable based on the 24-hour recall may be more accurate and less prone to recall bias [(see 42)], the variable based on a 7-day recall period is likely more stable and better captures an individual's habitual diet. In rural contexts where consumption varies daily because access to (and affordability of) food items cannot be guaranteed, a longer recall period may produce smoother results.

As is evident from Table 1, DDS for mothers and children varies with FAW infestation. The variation is more evident if we use the 24-hour rather than the 7 days recall period. The difference is larger and statistically significant for children's DDS in households with high-FAW intensity, while there is no difference for low-FAW intensity households. Comparable results are obtained for mothers' DDS, except that now the difference for both high- and low-FAW intensity are statistically significant. For the 7-day recall period, there is no statistically significant difference in DDSs for low- and high-FAW intensity households for both children and their mothers. Thus, the two measures appear to produce mixed signals regarding the relation between the intensity of FAW infestation and dietary diversity. It appears as if mothers prioritize their children over themselves when deciding about who eats what, and thus aim to shield their children from the worst effects of crop damage by FAW.

Kernel density plots of dietary diversity scores of children and their mothers are provided in Fig 1. They suggest a weak negative correlation between FAW intensity and dietary diversity, most clearly seen from the 24-hour recall data. Appendix Figures A2-3 in S2 Appendix provide kernel density plots of the main intermediate variables (maize yields, maize sales, and pesticide use) for the three sub-groups. Maize yields and sales are more directly related to FAW infestation than DDS, so these plots reveal a stronger association.

Although there is no official cut-off point below which diets are considered "insufficiently diverse", some studies use a threshold value for DDS of 4 for children and 5 for women—this is sometimes referred to as the so-called Minimum Dietary Diversity Score (MDD-W) ([18,43], Food and Agriculture Organisation [44]. Using the 24-hour recall period and this minimum dietary diversity score, a national assessment in Uganda found that only 27% of children between 6 and 23 months had a DDS that was sufficiently diverse [18]. Given our mean DDS for children and mothers (Table 1), it appears that lack of dietary diversity is less of a problem for our sample than for the national level. Food and Agriculture Organisation [41] also recommends using the mean DDS and computing proportions above or below the mean. For our sample, 52 and 54% of children and mothers consumed less than the mean DDS.

## Ethics statement

During data collection, informed consent was sought from all respondents before each interview. Each data collector read out the consent statement from the questionnaire in the local language to ensure that the respondent fully understood the

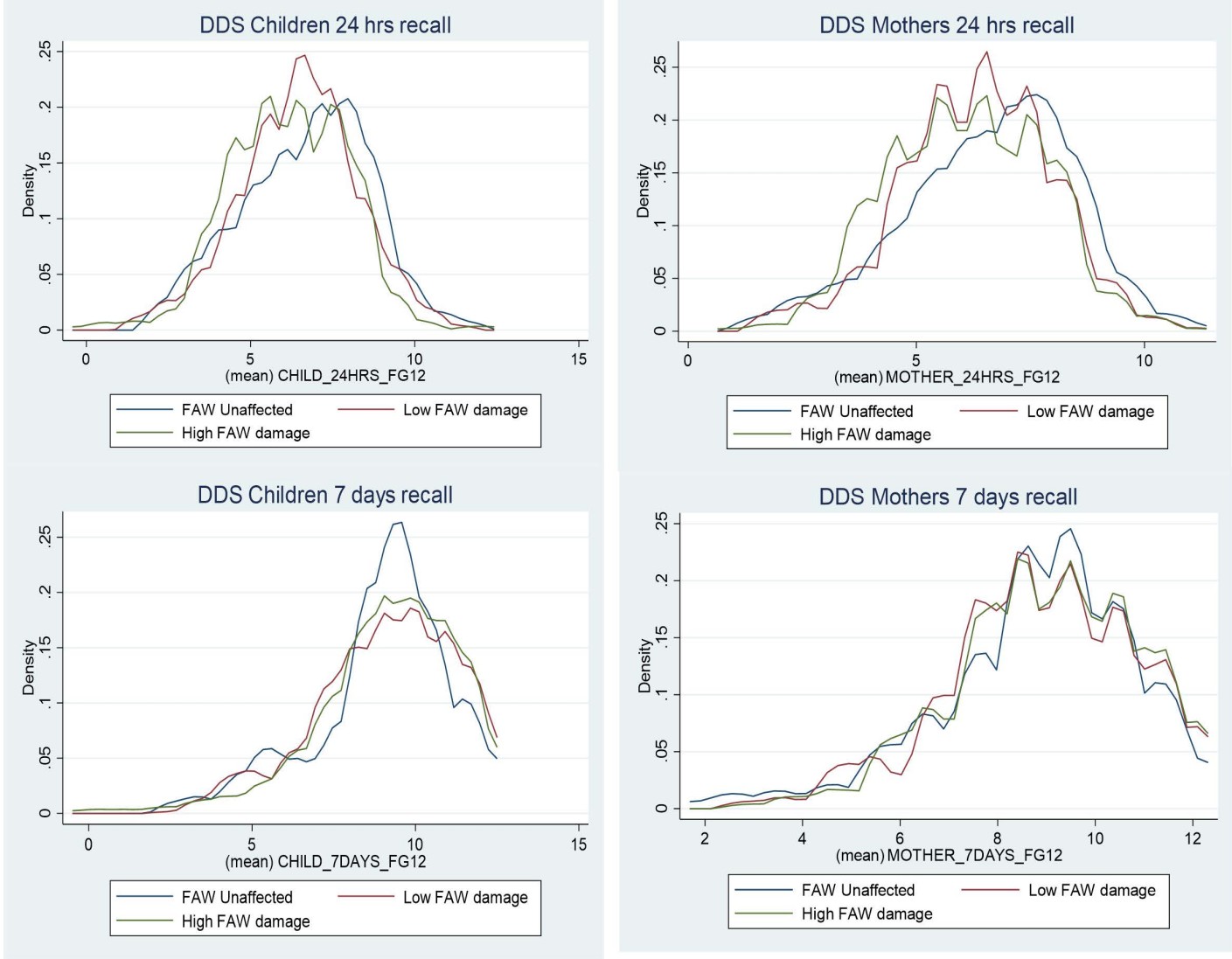

**Fig 1. Kernel density plots of DDS for children and their mothers by FAW Intensity (24 hours and 7 days recall periods).**

statement. The ethics statement was provided verbally and the respondent had the right to agree or not. The enumerator recorded yes or no in the questionnaire depending on the consent obtained from the respondent. The consent information is recorded in the questionnaires as part of the records. Details of the consent statement is provided in questionnaire included the supporting documents (S1 Questionnaire).

## Results and discussions

We estimated a multinomial logistic regression model to explain variation in FAW infestation (zero, low, high). We cluster standard errors at the household level to account for the interdependence of observations during the long and short rainy season. Full regression results are presented in Appendix Table A2. We distinguish between different types of explanatory variables: plot characteristics, household characteristics, farm management characteristics, Mundlak fixed effects,

a seasonal dummy, village fixed effects and our instrumental variables (Risk of FAW; distance between FAW infested plot and nearest affected plot; and distance between FAW infested plot and household's residence). Table 2 summarizes the marginal effect of the three IVs on the low- and high-FAW intensity regimes relative to "no FAW damage" (the base category).

As expected, increasing the share of FAW infested plots in the vicinity of a farmer is associated with an increase in the FAW intensity on that farmer's plot. However, this is only statistically significant for high FAW intensity. As expected, we also find a negative correlation between the distance between unaffected plots and the nearest FAW infested plot and FAW intensity. However, the correlation is not strong and is only statistically significant at high FAW intensity. With regards to the third instrument, we find a positive and statistically significant relationship between low FAW intensity and the distance from plot to residence. The instruments are significant determinants of FAW intensity either for low or high FAW intensity.

We then tested the validity of the instruments using the falsification method as summarised in Table 3. Since we had found statistical significance of the instruments on either low or high intensity, we now test whether the instrument is significant when regressed on the outcome variable for the sub-sample (FAW intensity=0). In all instances and for all outcome variables (except for one case), we find that the instruments are not statistically significant for the sub-sample FAW intensity=0. We therefore assume that the exclusion restriction holds and that the instruments are valid.

We further explore the validity of our instruments using partial F-statistics and the Durban-Wu-Hausman test (Table 4). From the partial F-statistics, the first stage regressions for the maize yield and insecticides-use regressions are above 10. The Durban-Wu-Hausman test results indicate that that the key explanatory variable is endogenous only for the maize sales outcome. Therefore, we prefer using MESR for the maize sales outcome and OLS for the models for the other outcome variables. However, we will estimate both sets of models for all dependent variables, to evaluate the robustness of our findings to alternative specifications.

**Table 2. Marginal Effects of Instruments on FAW Intensity.**

|  | (1) | (2) |
|---|---|---|
|  | Low FAW intensity | High FAW intensity |
| Risk of FAW (IV1) | 0.132 | 0.398*** |
|  | (0.080) | (0.080) |
| Distance between a FAW unaffected plot nearest affected plot (IV2) | −0.0001 (0.0005) | −0.001*** (0.001) |
| Ln Distance plot to homestead (IV3) | 0.034*** | −0.016 |
|  | (0.012) | (0.011) |
| Plot characteristics | √ | √ |
| Household characteristics | √ | √ |
| Farm management characteristics | √ | √ |
| Village fixed effects | √ | √ |
| Seasonal dummy | √ | √ |
| Mundlak fixed effects | √ | √ |
| Pseudo R2 | 0.171 | 0.171 |
| N | 1937 | 1937 |

*, **, *** significant at 10%, 5% and 1%; standard errors in parentheses; standard errors clustered at household level.

**Table 3. Instrument falsification tests.**

| | Outcome variables | | | | |
|---|---|---|---|---|---|
| | Ln(Maize yield) (kg/ha) (if FAW intensity = 0) | ihs_Maize sales (kg) (if FAW intensity = 0) | ihs_Insecticide use (liters/Ha) (if FAW intensity = 0) | DDS children (if FAW intensity = 0) | DDS mothers (if FAW intensity = 0) |
| HH characteristics | √ | √ | √ | √ | √ |
| Plot characteristics | √ | √ | √ | √ | √ |
| Mundlak effects | √ | √ | √ | √ | √ |
| Risk of FAW (IV1) | −0.130 (0.289) | −0.684 (1.427) | 0.021 (0.161) | 2.518 (2.034) | 2.985 (1.762) |
| Distance between a FAW unaffected plot nearest affected plot (IV2) | 0.0002 (0.0003) | −0.000 (0.001) | −0.000** (0.000 | −0.000 (0.001) | 0.000 (0.001 |
| Ln(distance) (IV3) | −0.010 (0.043) | 0.123 (0.181) | 0.004 (0.029) | −0.304 (0.178) | 0.197 (.134) |
| Village FE | √ | √ | √ | √ | √ |
| Constant | 7.771*** (0.491) | 8.822*** (2.096) | 0.131 (0.275) | 4.818** (2.220) | 4.118** (1.905) |
| Observations | 362 | 175 | 362 | 101 | 100 |
| R-squared | 0.115 | 0.227 | 0.05 | 0.439 | 0.556 |

*** p<0.01, ** p<0.05, * p<0.1; standard errors in parentheses; standard errors were clustered at household level

**Table 4. Partial F-Statistics and Durban-Wu-Hausman test from the first Stage Regression of the Two Stage Least Squares (2SLS).**

| | Outcome variables | | | | |
|---|---|---|---|---|---|
| | Ln(Maize yield) (kg/ha) | ihs_Maize sales (kg) | ihs_Insecticide use (liters/Ha) | DDS children | DDS mothers |
| HH characteristics | √ | √ | √ | √ | √ |
| Plot characteristics | √ | √ | √ | √ | √ |
| Mundlak effects | √ | √ | √ | √ | √ |
| Village FE | √ | √ | √ | √ | √ |
| Constant | 5.825 *** (0.240) | 7.114 *** (.967) | −0.162 (0.157) | 7.101 *** (0.814) | 7.368 (0.820) |
| Observations | 1937 | 1937 | 1,937 | 436 | 436 |
| R-squared | 0.179 | 0.204 | 0.174 | 0.227 | 0.275 |
| Partial F-Statistic | 12.16 | 7.94 | 12.45 | 7.87 | 7.87 |
| Durban-Wu-Hausman test (p-value) | 0.531 | 0.039 | 0.253 | 0.488 | 0.516 |

Durban-Wu-Hausman test was a post-estimation for the 2SLS regressions for each of the outcome variables. The Partial F-Statistic was based on the first stage regression from each of the 2SLS regressions.

### Impact of FAW intensity on maize yield, maize sales and insecticide-use

We evaluate how FAW infestation affects our intermediate variables, or the possible pathways to nutrition impacts. Following the test results for the IVs, we firstly present the results of FAW's impact on maize yield and insecticide-use based on OLS. We then present the results of FAW infestation's impact on maize sales using the MESR. Table 5 summarises the treatment effects of FAW intensity on maize yield from our OLS estimates. Our findings indicate that low FAW infestation does not have a statistically significant impact on maize yield. However, high FAW infestation has a negative and statistically significant effects on maize yield. At high FAW intensity, FAW caused maize yields to decline by 437 kg/Ha. This

**Table 5. Treatment effects of FAW intensity on maize yield based on OLS.**

| | Outcome = maize yield | |
|---|---|---|
| | **Low FAW** | **High FAW** |
| Average Treatment Effects | −163.17 (106.31) | −437.76*** (110.32) |
| Household characteristics | √ | √ |
| Plot characteristics | √ | √ |
| Mundlak Effects | √ | √ |
| Village FE | √ | √ |
| Constant | 1935.42*** (360.15) | 1681.60*** (303.10) |
| Observations | 1,171 | 1,171 |
| R-squared | 0.03 | 0.05 |

*, **, *** significant at 10%, 5%, and 1%; Standard errors were clustered at household level. Full regression results in appendix Table A3.

represents a 25.5% yield loss attributed to high FAW infestation given a mean yield of 1,714 kg/Ha (see Table 1) for plots that were unaffected by FAW.

As part of our robustness check, we additionally present the results of the MESR model for yield in Table 6 and the full regression of the second step regression in appendix Table A4. Our justification for presenting these results also is because we obtained a Partial F-statistic that was greater than 10 (Table 4). Moreover, the selection correction terms $\hat{\lambda}_{pij}$ are included in the full regression (appendix Table A4), enters significantly, thereby indicating the presence of selection bias—a further justification for the use of the MESR for the maize yield model. Table 6 indicates that the MESR produces comparable results with the OLS except that low FAW intensity now also has a significant impact on yield, and the effect size of high FAW intensity is lower than that obtained from OLS.

In Table 7, we summarize the impact of FAW intensity on maize sales based on the MESR model, which is our main result since FAW intensity in endogenous according to the Durban-Wu-Hausman test results. The ATT indicates how FAW intensity affects maize sales. The selection correction terms $\hat{\lambda}_{pij}$ are included in the full regression, which is presented in Appendix Tables A5. The coefficient of $\hat{\lambda}_{pij}$ enters significantly, which indicates the presence of selection bias—justifying the use of the MESR for maize sales. Both low and high FAW intensity has positive and statistically significant impacts on maize sales and the impact increases with intensity. Low and high FAW intensity caused maize sales to decline by 122 and 265 kg respectively. Considering the mean maize sales of 640 for unaffected households that we presented in Table 1, this represents maize sales declines of 19.05% and 41.38% caused by low and high FAW intensity respectively.

**Table 6. Treatment effects of FAW intensity on maize yield based on MESR.**

| | | | Predicted Counterfactual outcomes | | | Treatment Effect |
|---|---|---|---|---|---|---|
| **Outcome variable** | | **(1) Predicted actual outcomes** | **(2) FAW = 0** | **(3) FAW = 1** | **(4) FAW = 2** | **(5) ATT** |
| Maize yield | FAW = 0 | 1345 | – | 1185 | 1056 | |
| (kg/ha) | FAW = 1 | 1191 | 1332 | – | – | −141 *** |
| | FAW = 2 | 968 | 1280 | – | – | −313 *** |

*, **, *** significant at 10%, 5%, and 1%; ATT is the difference between the actual (column (1)) and counterfactual outcomes (column (2))

**Table 7. Treatment effects of FAW intensity on maize sales based on the MESR Model.**

| Outcome variables | | (1)<br>Predicted actual outcomes | Predicted Counterfactual outcomes | | | Treatment Effect |
| | | | (2)<br>FAW = 0 | (3)<br>FAW = 1 | (4)<br>FAW = 2 | (5)<br>ATT |
| --- | --- | --- | --- | --- | --- | --- |
| Maize sales | FAW = 0 | 487 | – | 410 | 382 | |
| (kg) | FAW = 1 | 287 | 389 | – | – | −122 *** |
| | FAW = 2 | 117 | 707 | – | – | −265 *** |

*, **, *** significant at 10%, 5%, and 1%; ATT is the difference between the actual (column (1)) and counterfactual outcomes (column (2)).

We also present the OLS results (Table 8) as a robustness check. The OLS impact estimates are negative and statistically significant but are higher than the ones from the MESR.

With regards to the impact of FAW intensity on insecticides-use, we present the OLS impact estimates as the main results in Table 9. We find a positive and statistically significant impact of low FAW infestation on insecticide-use. By contrast, we find no impact of high FAW intensity on insecticide use. Thus, low FAW intensity caused insecticide-use to increase 0.17 liters/hectare, which is a 40.5% increase over a mean insecticide-use of 0.42 liters/hectare for the unaffected plots.

We additionally provide MESR results for robustness in Table 10. This is because we obtained a partial F-statistics that is greater than 10 (Table 4) and at least one of the selection correction term $\hat{\lambda}_{pij}$ entered significantly in the second step MESR regression (appendix Table A6).

## Impact of FAW intensity on the dietary diversity for mothers and children

Following our discussion on the intermediary variables of the channel through which FAW infestation affects nutrition, we now turn to our main results. In Table 11, we present the impact of FAW infestation on the nutrition status of mothers and children, measured by DDS. Full regression results are presented in Appendix Table.A7. We find no statistically significant impact of low FAW infestation on the dietary diversity for both mothers and children, which suggests that low FAW infestation had no impact on the nutrition status of mothers and children. By contrast, high FAW infestation had a negative and statistically significant impact on the dietary diversity of both mothers and children. High FAW infestation caused DDS for mothers and children to decline by 0.89 and 0.57 respectively. Compared to the unaffected households, the decline in

**Table 8. Treatment effects of FAW intensity on maize sales based on OLS.**

| | Outcome = maize sales | |
| | Low FAW | High FAW |
| --- | --- | --- |
| Average Treatment Effects | −253.55 **<br>(116.61) | −401.61***<br>(123.02) |
| Household characteristics | √ | √ |
| Plot characteristics | √ | √ |
| Mundlak Effects | √ | √ |
| Village FE | √ | √ |
| Constant | 1024.52***<br>(373.09) | 919.06***<br>(326.84) |
| Observations | 599 | 575 |
| R-squared | 0.08 | 0.12 |

*, **, *** significant at 10%, 5%, and 1%; Standard errors were clustered at household level. Full regression results in appendix Table A3.

**Table 9. Treatment effects of FAW intensity on insecticide-use based on OLS.**

| | Outcome = insecticide-use | |
| | Low FAW | High FAW |
|---|---|---|
| Average Treatment Effects | 0.17** (0.08) | 0.10 (0.09) |
| Household characteristics | √ | √ |
| Plot characteristics | √ | √ |
| Mundlak Effects | √ | √ |
| Village FE | √ | √ |
| Constant | 0.16 (0.33) | 0.57* (0.33) |
| Observations | 1,171 | 1,171 |
| R-squared | 0.06 | 0.05 |

*, **, *** significant at 10%, 5%, and 1%; Standard errors were clustered at household level. Full regression results in appendix Table A3.

**Table 10. Treatment effects of FAW intensity on maize sales based on the MESR Model.**

| | | Predicted Counterfactual outcomes | | | | Treatment Effect |
|---|---|---|---|---|---|---|
| Outcome variables | | (1) Predicted actual outcomes | (2) FAW = 0 | (3) FAW = 1 | (4) FAW = 2 | (5) ATT |
| Insecticide use (ltr/ha) | FAW = 0 | 1.212 | – | 1.345 | 1.245 | |
| | FAW = 1 | 1.320 | 1.229 | – | – | 0.091*** |
| | FAW = 2 | 1.260 | 1.207 | – | – | 0.052*** |

*, **, *** significant at 10%, 5%, and 1%; ATT is the difference between the actual (column (1)) and counterfactual outcomes (column (2)).

**Table 11. Treatment effects of FAW on the DDS for children and mothers using OLS.**

| | Outcome = Child DDS | | Outcome = Mother DDS | |
| | Low FAW | High FAW | Low FAW | High FAW |
|---|---|---|---|---|
| Average Treatment Effects | −0.227 (0.300) | −0.566* (0.283) | −0.445 (0.278) | −0.891*** (0.273) |
| Household characteristics | √ | √ | √ | √ |
| Plot characteristics | √ | √ | √ | √ |
| Mundlak Effects | √ | √ | √ | √ |
| Village FE | √ | √ | √ | √ |
| Constant | 6.52*** | 7.37*** | 6.06*** | 7.65*** |
| | (0.93) | (0.77) | (0.88) | (0.78) |
| Observations | 240 | 273 | 237 | 272 |
| R-squared | 0.18 | 0.21 | 0.22 | 0.25 |

*, **, *** significant at 10%, 5%, and 1%; Standard errors were clustered at household level. Full regression results in appendix Table A3.

DDS for mothers and children represents a 13.2% and 6.3% respectively. Further, our results indicate a higher impact of high FAW infestation on mothers compared to their children.

In Table 12, we additionally present the MESR results for robustness because at least once of the selection correction terms $\hat{\lambda}_{pij}$ entered significantly in the second step MESR regression. Results from the MESR are comparable to those

**Table 12. Treatment effects of FAW on DDS for children and mothers based on the MESR.**

| Outcome variables | | Actual outcomes (1) | Predicted Counterfactual outcomes (2) FAW = 0 | (3) FAW = 1 | (4) FAW = 2 | Treatment Effects (5) ATT |
|---|---|---|---|---|---|---|
| Child DDS | FAW = 0 | 7.209 | – | 6.818 | 6.330 | |
| | FAW = 1 | 6.590 | 7.497 | – | – | −0.906*** |
| | FAW = 2 | 6.589 | 7.626 | – | – | −1.037*** |
| Mother DDS | FAW = 0 | 7.314 | – | 6.770 | 6.330 | |
| | FAW = 1 | 6.665 | 7.779 | – | – | −1.113*** |
| | FAW = 2 | 6.677 | 7.967 | – | – | −1.290*** |

*, **, *** significant at 10%, 5%, and 1%; ATT is the difference between the actual (column (1)) and counterfactual outcomes (column (2)).

obtained from the OLS except that low FAW intensity has no significant impact on DDS for mothers and children. Further, the effect size of high FAW intensity from the MESR is larger for both mother and children.

The non-experimental nature of FAW diffusion across the African continent implies that causal impact statements rest on assumptions. To further check the robustness of our findings we also use the Control Function (CF) Approach. The CF approach incorporates instruments in its set up and proceeds in two steps, similar to the MESR approach. Using a flexible form of the CF, where we use a fractional treatment offers an alternative robustness check to the MESR and OLS. Using a fractional probit in the first stage of the MESR is not possible. In the first stage of the CF, we explain variation in FAW intensity as a reduced form regression (including our instrumental variables risk of FAW; distance between FAW infested plot and nearest affected plot; and distance between FAW infested plot and household's residence) and exogenous variables as regressors), from which we derive "generalized residuals". We then estimate the second stage as a linear regression model, regressing our outcome variables on the treatment variable, exogenous controls, and the derived generalized residuals (see Wooldridge 2015). We now use a fractional treatment variable, so that FAW intensity is treated as a continuous share ranging from 0 to 1. We present results of the second stage in Table 13, and they are comparable to the ones reported earlier for the OLS and MESR. We find that farmers suffering from more intense pest pressure harvest and sell smaller quantities of maize, insecticide use is unaffected (across the entire range of infestation levels), and there is a significant negative effect of FAW infestation on dietary diversity of children and mothers. Thus our findings on the impact of FAW on dietary diversity is robust to the different model specifications.

**Table 13. Robustness checks using the Control Function Approach.**

| Model | Outcome Variable (1) | FAW Intensity (Fractional Treatment) (2) |
|---|---|---|
| Control Function using fractional probit in the first stage ("risk of FAW; distance between FAW infested plot and nearest affected plot; and distance between FAW infested plot and household's residence" as IV) | Yield | −676.14*** |
| | Sales | −293.55*** |
| | Insecticide-use | 0.214 |
| | DDS Children | −0.966*** |
| | DDS Mothers | −1.276*** |

*, **, *** significant at 10%, 5%, and 1%; Models 3 and 4 were estimated manually; hence we bootstrapped the standard errors in the second stage regression. Full regressions can be made available upon request.

## Conclusions

We study the impact of fall armyworm (FAW) infestations on nutrition security outcomes in eastern Uganda, measured by dietary diversity scores of the most vulnerable household members—children under 5 years and mothers. We also studied FAW's impact pathway to nutrition status and asked whether FAW's impact on nutrition security is caused by reduced maize yields and sales or increased expenditures associated with insecticide use.

We find that FAW infestation causes maize yields and sales to fall with appreciably large effect sizes. FAW infestation does not affect maize yield at low FAW intensity but causes substantial yield losses of 25% at high intensity. FAW infestation has more adverse effects on maize sales even at low intensity. This is consistent with households consuming their own crop and selling surpluses, so sales are relatively more volatile than own consumption or even yields. High and low FAW infestation causes maize sales to decline by 19% and 41% respectively. These findings are consistent with previous research and suggest a plausible channel from FAW infestation to reduced nutrition security. Our findings reveal more nuanced effects of FAW intensity on insecticide-use. While insecticide-use significantly increases with low FAW infestation, we find no impact of high FAW on insecticide use. This may reflect (the anticipation of) binding cash and credit constraints, but farmers may also rationally choose to use less pesticides on plots where they feel they cannot salvage any crops.

Our results on FAW's impact on dietary diversity show that FAW causes significant and negative effects on the diets of mothers and children—the most vulnerable household members. Low FAW infestation rates have no effect on dietary diversity for both mothers and children, while we find that high FAW intensity caused declines in the dietary diversity of mothers and children. Dietary diversity of children under five declines by 6.3% while the dietary diversity for mothers falls by 13.2%. Our finding that the impact is only found at high FAW intensity, also signifies that household diets are rather resilient to FAW at low exposure. In light of the precarious baseline nutrition condition of this sample population, nevertheless, these are undesirable nutrition outcomes

## Supporting information

**S1 Appendix. Appendix tables.**
(DOCX)

**S2 Appendix. Appendix figures.**
(DOCX)

**S1 File. Inclusivity in global research questionnaire.**
(DOCX)

**S1 Questionnaire. Questionnaire used for field data collection.**
(DOCX)

## Acknowledgments

The authors are grateful to *icipe* and Makerere University, College of Agricultural and Environmental Sciences for managing and coordinating the data collection process as well as the communities from whom the data was collected for their cooperation.

## Author contributions

**Conceptualization:** Brian Chisanga, Menale Kassie.

**Data curation:** Brian Chisanga.

**Formal analysis:** Brian Chisanga.

**Funding acquisition:** Menale Kassie.

**Methodology:** Brian Chisanga, Menale Kassie.

**Project administration:** Menale Kassie.

**Software:** Brian Chisanga.

**Supervision:** Menale Kassie.

**Writing – original draft:** Brian Chisanga.

**Writing – review & editing:** Menale Kassie.

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
