## [Editor Report · Decision Letter 0]

9 Aug 2025

PONE-D-25-38658Fall Armyworm Infestation, Maize production and Nutrition Security Evidence from UgandaPLOS ONE

Dear Dr.  Chisanga,

Thank you for submitting your manuscript to PLOS ONE. After careful consideration, we feel that it has merit but does not fully meet PLOS ONE’s publication criteria as it currently stands. Therefore, we invite you to submit a revised version of the manuscript that addresses the points raised during the review process.

We look forward to receiving your revised manuscript.

Kind regards,

Rodney N. Nagoshi, Ph.D.

Academic Editor

PLOS ONE

Journal Requirements:

“financial support for this research by the following organizations and agencies: European Union (DCI-FOOD/2018/402-634)”

7. We note that Figure A2 in your submission contain map images which may be copyrighted. All PLOS content is published under the Creative Commons Attribution License (CC BY 4.0), which means that the manuscript, images, and Supporting Information files will be freely available online, and any third party is permitted to access, download, copy, distribute, and use these materials in any way, even commercially, with proper attribution. For these reasons, we cannot publish previously copyrighted maps or satellite images created using proprietary data, such as Google software (Google Maps, Street View, and Earth). For more information, see our copyright guidelines: http://journals.plos.org/plosone/s/licenses-and-copyright.

1. You may seek permission from the original copyright holder of Figure A2 to publish the content specifically under the CC BY 4.0 license. 

Additional Editor Comments:

Your manuscript is being returned without review because it does not meet the formatting requirements for PLOS One. Please see guidelines listed in detail at https://journals.plos.org/plosone/s/submission-guidelines. Specifically, please organize the paper with a discrete Methods section that is distinct from the description of the results. The figures and tables in the results section should be limited to those describing the experimental results. Tables/figures that are only used to validate the methods used should be moved to the Methods section or Supplement.

Upon completion of these revisions, the manuscript may be resubmitted where it will then proceed through the normal review process.

---

## [Author Response · Author response to Decision Letter 1]

9 Oct 2025

Comment 1: Please ensure that your manuscript meets PLOS ONE's style requirements, including those for file naming. The PLOS ONE style templates can be found at

Response 1: The manuscript has been significantly revised and re-formatted in line with the journal requirements. The revised /manuscript has been re-submitted through the PLOS ONE online system

Comment 2: Please include a complete copy of PLOS’ questionnaire on inclusivity in global research in your revised manuscript. Our policy for research in this area aims to improve transparency in the reporting of research performed outside of researchers’ own country or community. The policy applies to researchers who have travelled to a different country to conduct research, research with Indigenous populations or their lands, and research on cultural artefacts. The questionnaire can also be requested at the journal’s discretion for any other submissions, even if these conditions are not met. Please find more information on the policy and a link to download a blank copy of the questionnaire here: https://journals.plos.org/plosone/s/best-practices-in-research-reporting. Please upload a completed version of your questionnaire as Supporting Information when you resubmit your manuscript.

Response 2: We have included the filled-in copy of PLOS’ questionnaire on inclusivity in global research as part of the supporting documents. We have also included our data collection questionnaire in the supporting documents.

Comment 3: Please provide additional details regarding participant consent. In the ethics statement in the Methods and online submission information, please ensure that you have specified (1) whether consent was informed and (2) what type you obtained (for instance, written or verbal, and if verbal, how it was documented and witnessed). If your study included minors, state whether you obtained consent from parents or guardians. If the need for consent was waived by the ethics committee, please include this information.

Response 3: We have stated that consent was obtained verbally from respondents. A statement was included in the questionnaire and read out by enumerators to each of the respondents before the interview proceeds.

Comment 4: We note that the grant information you provided in the ‘Funding Information’ and ‘Financial Disclosure’ sections do not match.

Response 4: Thanks. The correct grant information is: European Union (DCI-FOOD/2018/402-634)

Comment 5: Thank you for stating the following financial disclosure:

“financial support for this research by the following organizations and agencies: European Union (DCI-FOOD/2018/402-634)”

Response 5: Yes, the funders only supported the study financially. “Funders had no role in study design, data collection and analysis, decision to publish, or preparation of the manuscript”

Comment 6: We note that Figure A2 in your submission contain map images which may be copyrighted. All PLOS content is published under the Creative Commons Attribution License (CC BY 4.0), which means that the manuscript, images, and Supporting Information files will be freely available online, and any third party is permitted to access, download, copy, distribute, and use these materials in any way, even commercially, with proper attribution. For these reasons, we cannot publish previously copyrighted maps or satellite images created using proprietary data, such as Google software (Google Maps, Street View, and Earth). For more information, see our copyright guidelines: http://journals.plos.org/plosone/s/licenses-and-copyright.

Response 6: I have removed the figure from the submission

---

## [Editor Report · Decision Letter 1]

30 Oct 2025

Fall Armyworm Infestation, Maize production and Nutrition Security Evidence from Uganda

PONE-D-25-38658R1

Dear Dr. Chisanga,

We’re pleased to inform you that your manuscript has been judged scientifically suitable for publication and will be formally accepted for publication once it meets all outstanding technical requirements.

Kind regards,

Zhi-Yao He, Ph. D.

Academic Editor

PLOS ONE
---

## [Editor Report · Acceptance letter]

PONE-D-25-38658R1

PLOS ONE

Dear Dr. Chisanga,

I'm pleased to inform you that your manuscript has been deemed suitable for publication in PLOS ONE. Congratulations! Your manuscript is now being handed over to our production team.

Kind regards,

on behalf of

Dr. Zhi-Yao He

Academic Editor

PLOS ONE